# Mechanisms of PKC-Mediated Enhancement of HIF-1α Activity and its Inhibition by Vitamin K2 in Hepatocellular Carcinoma Cells

**DOI:** 10.3390/ijms20051022

**Published:** 2019-02-26

**Authors:** Jinghe Xia, Iwata Ozaki, Sachiko Matsuhashi, Takuya Kuwashiro, Hirokazu Takahashi, Keizo Anzai, Toshihiko Mizuta

**Affiliations:** 1Department of Internal Medicine, Saga Medical School, Saga University, 5-1-1 Nabeshima, Saga 849-8501, Japan; xiajinghe@hotmail.com (J.X.); matsuha2@edu.cc.saga-u.ac.jp (S.M.); hepatology0321@gmail.com (T.K.); hepatology28@msn.com (H.T.); akeizo0479@gmail.com (K.A.); 2Health Administration Center, Saga Medical School, Saga University, 5-1-1 Nabeshima, Saga 849-8501, Japan; 3Department of Internal Medicine, Fujikawa Hospital, 1-2-6 Matsubara, Saga 840-0831, Japan; mizuta1221@gmail.com

**Keywords:** Vitamin K2, Hypoxia-inducible factor-1α (HIF-1α), Protein kinase C (PKC), Hepatocellular carcinoma cells

## Abstract

Hypoxia-inducible factor 1 (HIF-1) plays important roles in cancer cell biology. HIF-1α is reportedly activated by several factors, including protein kinase C (PKC), in addition to hypoxia. We investigated the role of PKC isoforms and the effects of vitamin K2 (VK2) in the activation process of HIF-1α. Human hepatocellular carcinoma (HCC)-derived Huh7 cells were cultured under normoxic and hypoxic (1% O_2_) conditions with or without the PKC stimulator TPA. The expression, transcriptional activity and nuclear translocation of HIF-1α were examined under treatment with PKC inhibitors, siRNAs against each PKC isoform and VK2. Hypoxia increased the expression and activity of HIF-1α. TPA increased the HIF-1α activity several times under both normoxic and hypoxic conditions. PKC-δ siRNA-mediated knockdown, PKC-δ inhibitor (rottlerin) and pan-PKC inhibitor (Ro-31-8425) suppressed the expression and transcriptional activity of HIF-1α. VK2 significantly inhibited the TPA-induced HIF-1α transcriptional activity and suppressed the expression and nuclear translocation of HIF-1α induced by TPA without altering the HIF-1α mRNA levels. These data indicate that PKC-δ enhances the HIF-1α transcriptional activity by increasing the nuclear translocation, and that VK2 might suppress the HIF-1α activation through the inhibition of PKC in HCC cells.

## 1. Introduction

Hypoxia is a common microenvironment observed in a broad range of diseases, particularly solid tumors, including hepatocellular carcinoma (HCC) [1,2]. To maintain oxygen homeostasis, higher-level eukaryotes have developed specialized factors and mechanisms to adapt to hypoxic conditions. In mammalian cells, hypoxia-inducible factor 1 (HIF-1) plays critical roles in cellular and systemic oxygen maintenance and functions as a master transcription factor by binding to the hypoxia responsive element (HRE) of targeted genes, such as vascular endothelial growth factor (VEGF) and erythropoietin (EPO) [3,4]. Consequentially, oxygen availability is increased by promoting angiogenesis and erythropoiesis. HIF-1 is a heterodimeric factor consisting of α and β subunits that share the same basic helix-loop-helix construction. The β subunit is expressed constitutively, while the α subunit shows remarkably varied expression and activity in response to the environment.

In normal atmosphere (normoxic conditions), HIF-1α is hydroxylated by prolyl-hydroxylases (PHDs) at proline residues in an oxygen-dependent manner, allowing for the recognition and ubiquitination by the Von Hippel–Lindau (VHL) E3 ubiquitin ligase, which substantially labels them for rapid degradation by the proteasome [5]. Under hypoxic condition, HIF-1α is stabilized, followed by an increase in translocation to the nucleus and transcriptional activity when the hydroxylation is inhibited. However, stabilization of HIF-1α can occurs even under normoxic conditions when VHL is lost due to somatic mutations or epigenetic changes [6]. Furthermore, the HIF-1α expression and its stabilization are also regulated through other mechanisms. NF-κB is reported to be activated under hypoxic conditions and modulates the HIF-1α mRNA expression. The cross-talk between these two factors has also been observed in several studies [7,8].

Protein kinase C (PKC) was investigated by several groups as a potential regulator of HIF-1 activity. Datt et al. reported that PKC-ζ transactivates HIF-1α by promoting its association with p300 in renal cancer [9], and Page et al. showed that PKCs play an important role in increasing the HIF-1α gene transcription [10]. Several studies have revealed that PKC-δ is activated by hypoxia and subsequently increases the HIF-1α stability in human cancer cells [11,12,13], and PKC-α seems to be involved in the activation of HIF-1α along with cAMP response element binding protein (CREB) in rat tissue [14,15]. 

We previously showed that TPA-activated PKCs were involved in the activation of NF-κB, and vitamin K2 (VK2) suppressed the NF-κB activity by suppressing the PKC activity [16]. In the present study, we investigated the role of PKC isoforms in the activation of HIF-1α in human HCC cells and the effects of VK2 on HIF-1α activation.

## 2. Results

### 2.1. PKC-δ—But not PKC-α or PKC-ε—Is Involved in the Activation of HIF-1α and Inhibition by VK2 Under Both Normoxic and Hypoxic Conditions

To investigate the effects of VK2 and PKCs on the HIF-1α transcription activity in Huh7 cells, a HIF-1α reporter luciferase plasmid containing the HRE was employed. As shown in Figure 1A, compared with normoxic conditions, the transcription activity of HIF-1α was 10 times higher under hypoxic conditions. In addition, the PKC activator TPA induced three-fold greater HIF-1α activity under both normoxic and hypoxic conditions compared to the control. Similarly, VK2 abrogated the TPA-induced HIF-1α transcription activity in a dose-dependent manner under both normoxic and hypoxic conditions. The ethanol did not show any significant effect on the activity, indicating that PKCs were involved in the HIF-1α activation and that VK2 might suppress the HIF-1α activity via the PKCs in Huh7 cells.

To clarify which PKCs were involved in stimulating the HIF-1α transcription activity, we performed siRNA-mediated PKC knockdown in Huh7 cells. The specific siRNAs of PKC-α (cPKC), PKC-δ (nPKC) and PKC-ε (nPKC) were used as previously described [16]. As shown in Figure 1B, knockdown of PKC-δ significantly suppressed the TPA-induced HIF-1α activity, but little effect on the hypoxia-induced HIF-1α activity without TPA was noted. However, we detected no marked changes in the HIF-1α activity after the knockdown of PKC-α or PKC-ε either in hypoxia or in the TPA induced level. We next used specific inhibitors of PKCs as indicated in Figure 1C to confirm the involvement of PKCs. The PKC-δ inhibitor rottlerin (10 nM) significantly inhibited the TPA-induced HIF-1α transcriptional activity to the same degree as the pan-PKC inhibitor Ro-31-8425 (100 nM). However, the PKC-α inhibitor Gö6976 (10 nM) exerted no marked effects on the HIF-1α activity. These results suggested that PKC-δ participated in the activation of HIF-1α transcription activity in Huh7 cells.

### 2.2. PKC-δ is Involved in the HIF-1α Protein Expression, and VK2 Suppresses the TPA-Induced HIF-1α Protein Expression

The luciferase assay above revealed the involvement of PKC-δ in the activation of HIF-1α transcriptional activity in Huh7 cells. We assessed the HIF-1α expression after hypoxia by Western blotting. As shown in Figure 2A,B, HIF-1α protein accumulated in Huh7 cells under hypoxia exposure, while the expression of HIF-1α mRNA remained unchanged. The mRNA expression of vascular endothelial growth factor (VEGF), one of the downstream responsible targets of HIF-1α, was stimulated, as expected. However, on examining the activation of PKC-δ by detecting phosphorylated PKC-δ and the total PKC-δ expression, no clear increase in PKC-δ expression or activation was observed after hypoxia exposure.

We next performed a Western blotting analysis using specific siRNAs against various PKC isoforms. As shown in Figure 2C,D, after 24-h treatment with 50 nM TPA under hypoxic conditions, the HIF-1α expression was upregulated. In contrast, knockdown of PKC-δ inhibited the expression of HIF-1α under hypoxic conditions, irrespective of TPA induction.

Experiments concerning the effect of VK2 on the HIF-1α expression were performed under hypoxic conditions both with and without TPA induction in Huh7 cells. As shown in Figure 2D, VK2 suppressed the HIF-1α expression induced by TPA in a dose-dependent manner under hypoxic conditions in Huh7 cells, while no marked effect was observed under hypoxic conditions without TPA stimulation. We also investigated the effects of TPA and PKC isoforms on the HIF-1α mRNA level in Huh7 cells, but no significant changes in the HIF-1α mRNA expression were observed (Figure 2E, left and middle panel). Similarly, VK2 showed no significant effects on the HIF-1α mRNA expression, suggesting that the PKC-dependent control of the HIF-1α expression and transcriptional activation is regulated by posttranscriptional levels.

### 2.3. PKC-δ Regulates the TPA-Induced Recruitment of HIF-1α, and VK2 Abrogates the Induction of HIF-1α Recruitment by TPA in Huh7 Cells

To assess the role of PKCs in the activation of HIF-1α and the effect of VK2 in Huh7 cells, we performed a ChIP assay under hypoxic conditions with and without TPA in Huh7 cells. As shown in Figure 3A, after TPA induction, the recruitment of HIF-1α to the VEGF promoter was enhanced. In PKC siRNA-mediated knockdown experiments, we found that knockdown of PKC-δ decreased the HIF-1α recruitment induced by TPA, with little effect observed on the hypoxia-induced HIF-1α recruitment activity without TPA. Consistent with the luciferase assay results, as shown in Figure 3B, VK2 abrogated the recruitment of HIF-1α induced by TPA in a dose-dependent manner under hypoxic conditions and inhibited the hypoxia-induced recruitment of HIF-1α. These results from different approaches strongly support the critical role of PKC-δ in the TPA-activated HIF-1α transcriptional activation, and suggest that the suppressive effect of VK2 might be mediated by PKC-δ in Huh7 cells.

### 2.4. PKC-δ Knockdown and VK2 Suppresses the Nuclear Translocation of HIF-1α Stimulated by TPA

Since the results above suggested that PKC-δ and VK2 regulate the HIF-1α transcriptional activity through the regulation of the HIF-1α protein expression and its recruitment to HIF-1α responsive elements in Huh7 cells, we examined the nuclear translocation of HIF-1α in order to investigate the mechanisms underlying the regulation of HIF-1α activity in greater detail.

We first performed a cell fluorescence microscope assay. Pictures for each experiment were taken using the same fields of cells reflecting the HIF1α-GFP expression in the nucleus. As shown in Figure 4A, the pTK-EGFP-N1 plasmid was expressed, and the Hoechst nuclear staining of Huh7 is shown in the upper part of the panels. Under normoxic conditions, the pTK-HIF-1α-EGFP-N1 expression was extremely low. After 24 h exposure to hypoxic conditions, the expression of pTK-HIF-1α-EGFP-N1 was increased, and TPA further enhanced the nuclear expression under hypoxic conditions, as shown in the lower panels of Figure 4A. As summarized in Figure 4B, knockdown of PKC-δ significantly inhibited the increased nuclear translocation of HIF-1α in Huh7 cells, while knockdown of PKC-α and PKC-ε showed no marked inhibitory effect on the translocation. VK2 significantly inhibited the nuclear expression of HIF-1α induced by hypoxia as well as the TPA-enhanced HIF-1α nuclear translocation in Huh7 cells, as shown in Figure 4C.

To confirm the nuclear translocation of HIF-1α, we performed a Western blotting analysis of the nuclear extracts of Huh7 cells. As shown in Figure 5A, TPA enhanced the nuclear expression of HIF-1α in Huh7 cells. This increase in expression was clearly suppressed by the knockdown of PKC-δ but not by that of PKC-α or PKC-ε. A similar suppression pattern with VK2 treatment was observed, as shown in Figure 5B, which is consistent with the results obtained from the luciferase and fluorescence microscope assays.

## 3. Discussion

In this study, we showed that the tumor promoter TPA increased the HIF-1α transcriptional activity in a PKC-δ-dependent manner, and VK2 suppressed the TPA-stimulated HIF-1α transcriptional activity in a human HCC cell line. A putative signaling pathway obtained from the study is summarized in Figure 6.

TPA is a potent stimulator of PKCs, and the involvement of PKCs in HIF-1 activation has been reported [9,10,11,12,13,14,15]. Among PKC isoforms, our results demonstrated the critical role of PKC-δ in the activation process of TPA-induced HIF-1α, consistent with the findings of previous reports [11,12,13]. Interestingly, a similar TPA-mediated increase in the HIF-1α transcriptional activity was observed under both normoxic and hypoxic conditions, suggesting that the PKC-δ-mediated increase in the HIF-1α transcriptional activity includes a hypoxia-independent mechanism. We noted that TPA stimulated the HIF-1α transcriptional activity without increasing its mRNA expression, consistent with the findings of previous reports showing that the stability of HIF-1α protein was increased by PKC-δ [11]. Furthermore, our siRNA-mediated knockdown of PKC isoforms showed that PKC-δ played critical roles in the nuclear translocation and recruitment of HIF-1α to the promoter region of the target gene in addition to maintaining its protein expression.

In this study, we found that VK2 inhibited the TPA-stimulated HIF-1α transcriptional activity under both normoxic and hypoxic conditions. Although HIF-1 complex is regulated by hypoxia, a variety of inflammatory mediators and microbial components are shown to upregulate HIF-1 even under normoxic conditions. Indeed, non-hypoxic stimuli, including inflammatory mediators such as thrombin [17], TNF-α [18], LPS [19] and growth factors [20,21,22], have been proven to have functional effects on HIF-1 activity. Among the mechanisms underlying the non-hypoxic stimuli-mediated activation of HIF-1, NF-κB performs important functions by stimulating the promoter of the HIF gene [8,13,23,24] and through its involvement in the cross-talk between the NF-κB pathway and HIF pathway that has been documented at the protein level in response to TNF-α [18] or hepatocyte growth factor (HGF) [22]. We and others [16,25,26] have previously shown that VK2 inhibits NF-κB activation through the inhibition of PKC activity, so the suppression of TPA-stimulated HIF-1 activation by VK2 might be mediated by PKCs and/or NF-κB. We did not observe any significant change in the HIF-1α mRNA levels by TPA, PKC siRNAs or VK2 in Huh7 cells; therefore, TPA-induced PKCs and/or NF-κB activation may not play critical roles in the transcription of the HIF-1α gene in Huh7 cells but instead contribute to HIF-protein stabilization, nuclear translocation and recruitment to the promoter region of HIF target genes.

Hypoxia is a common characteristic phenomenon in solid tumors, including HCC [1,2]. HCC is a chemotherapy-resistant malignant tumor, and several reagents, such as sorafenib, are currently approved for the systemic treatment of advanced HCC (reviewed in reference [27]). Sorafenib has been shown to extend the survival period of end-stage HCC patients by several months. However, the effect of chemotherapeutic reagents on HCC is not yet satisfactory. While sorafenib has been shown to inhibit the HIF-1α protein expression [28], an elevated level of HIF-1 is reportedly associated with an increased likelihood of developing resistance to sorafenib in HCC [29,30]. Relative long exposure to hypoxia can cause tumor progression, local invasion, phenotype alteration and therapeutic resistance [29,30,31]. HIF-1 transcription factor plays a critical role in the adaption process and may increase the likelihood of developing chemotherapeutic resistance, therefore, finding a way to control the HIF-1 activity is important for enhancing the therapeutic effects of cancer.

In addition to HIF-1, NF-κB is also involved in the development of chemoresistance to anti-cancer reagents in many tumors [32,33]; therefore, simultaneous targeting of HIF and NF-κB might be the ideal approach to treating chemoresistant cancers. Taken together, our present and previous findings have shown that VK2 suppresses the NF-κB activation and HIF-1 activation through the modulation of PKC activity, probably in an isoform-specific manner. VK2 and its derivatives might thus be potential pharmacological candidates for the simultaneous targeting of NF-κB and HIF.

Abnormal des-carboxy prothrombin (DCP) was specifically detected in patients with HCC and used as a specific diagnostic marker of HCC, while also being a useful prognostic marker [34,35]. Since in vitro studies [36,37,38] showed the anti-proliferative effects of VK on several cancer cells including HCC and reduced the DCP production, it was expected that VK2 might show anti-tumor effects in HCC patients. In the clinical settings, we and others previously reported a VK2 analog, menaquinone (MK-4) which was used in the current in vitro study suppressed the HCC development or recurrence after curative therapy [39,40,41]. Although these initial studies demonstrated the favorable effects of MK-4 on the recurrence of HCC and meta-analysis of RCTs [42] showed the favorable effects on recurrence of HCC, a large randomized control trial failed to show the advantage of MK-4 administration in inhibiting the recurrence of HCC after curative treatment [43], therefore the effects of vitamin K2 on liver cancer are limited and remain controversial.

After VK was originally discovered as a cofactor of functional coagulation production in the liver, it was also shown that VK had beneficial effects on extrahepatic tissue including bone and cardiovascular health. Recent studies used MK-7, another subtype of VK2 analog, and showed that MK-7 supplementation even in low dose contributed to keep optimal extrahepatic VK status improving circulating concentration of uncarboxylated osteocalcin (ucOC) and/or dephopho-uncarboxylated matrix Gla protein (dp-uc MGP) [44,45,46]. On the other hand, high vitamin MK-7 intake supports the reduction of body weight, abdominal fat and visceral fat [47]. However, these reports were affected by the limitations of study design [48], therefore further studies are required to confirm the effects of VK on human health as well as experimental investigation to clarify the mechanism of VK.

## 4. Materials and Methods

### 4.1. Cell Culture, Hypoxic Conditions and Reagents

The Huh7 cell line was obtained from the Japanese Cancer Research Resources Bank (Osaka, Japan). Cells were cultured and maintained in Dulbecco′s modified Eagle′s medium (DMEM) (Sigma-Aldrich, St. Louis, MO, USA) containing 10% fetal bovine serum (FBS) and 1% penicillin and streptomycin (Life Technologies, Carlsbad, CA, USA) in 5% CO_2_ at 37 °C. For the hypoxic condition, cells were incubated at 5% CO_2_ with 1% O_2_ level balanced with N_2_ in a hypoxic chamber (O_2_/CO_2_ incubator 9000E; WAKENYAKU, Kyoto, Japan) at 37 °C. Menatetrenone, a vitamin K2 analogue, was provided by Eisai Co (Tokyo, Japan). VK2 was dissolved in ethanol and stocked at 1 mM concentration and stock solution were diluted to the final concentration as indicated. Cells were treated with ethanol as control or the indicated concentration of VK2. The PKC inhibitors Ro-31-8425, Gö6976 and rottlerin were from Calbiochem (San Diego, CA, USA), and 12-O-tetradecanoylphorbor-13-acetate (TPA) was obtained from Sigma-Aldrich (St. Louis, MO, USA). SiRNAs against PKC-α, PKC-δ and PKC-ε isoforms (HP validated, S1003007308, S102660539, S100587784) and Allstar negative control siRNA (S1027281) were obtained from Qiagen (Heiden, Germany). Anti-HIF-1α and anti-PKC-δ (S505) antibodies were obtained from Cell Signaling Technology (Beverly, MA, USA), and anti-PKC-δ was obtained from Santa Cruz Biotechnology (Santa Cruz, CA, USA). Anti-human β-actin antibody was obtained from Biomedical Technologies (Stoughton, MA, USA).

### 4.2. Luciferase Reporter Gene Assay

The HIF-1α transcriptional activity was detected by a luciferase assay performed according to the method described by the supplier (Promega, Madison, WI, USA). Huh7 cells were transfected with HIF-1α reporter plasmid after 3–4 days of culture with DMEM containing 10% FBS. Transfected cells were seeded onto 48-well plates at 2 × 10^3^ cells per well in DMEM with 10% FBS and incubated until 80% confluent at 37 °C before use. After being treated with different concentrations of VK2 or PKC isoform-specific inhibitors or knockdown with PKC-specific siRNA under hypoxic/normoxic conditions in the presence/absence of 50 nM TPA/DMSO for 24 h, the cells were washed twice with PBS and carefully lysed in 1× passive lysis buffer (Promega). Cell extracts were immediately assayed for their luciferase activity using a Berthold Luminometer (MLR-100 Micro Lumino Reader; Corona Electric, Ibaraki, Japan). The amount of protein in the cell extracts was determined through normalization with a Pierce 660nm protein assay system (Thermo Scientific, Carlsbad, CA, USA).

### 4.3. Real-Time qRT-PCR, Real-Time Quantitative Reverse Transcription Polymerase Chain Reaction

Total RNA was isolated from cells using RNAiso Plus (Takara, Kusatsu, Japan) and reverse transcribed to cDNA using a High Capacity cDNA Reverse Transcription Kit (Thermo Fisher, Waltham, MA, USA) according to the manufacturer’s instructions. Real-time PCR using SYBR Select Master Mix (Thermo Fisher) was performed on a StepOnePlus system (Applied Biosystems) according to the manufacturer’s instructions. The following primer pairs were used to detect the mRNAs: HIF-1α (forward 5′-GTTCGCATGTTGATAAGGC-3′ and reverse 5′-GGAGAAAATCAAGTCGTGC-3′), VEGF (5′-TGCACCCACGACAGAAGGGGA-3′ and 5′-TCACCGCCTTGGCTTGTCACAT-3′) and GAPDH (5′-CCACCCATGGCAAATTCCATGGCA-3′ and 5′-TCTAGACGGCAGGTCAGGTCCACC-3′). Data were analyzed using the comparative Ct (ΔΔCt) method, and the expression of target genes was normalized to GAPDH. Each experiment was performed in triplicate.

### 4.4. Western Blotting

Cells cultured under various conditions were collected and lysed with SDS buffer (50 mM Tris (pH 6.8), 2.3% SDS and 1 mM PMSF). The cell debris was eliminated by centrifugation at 12,000× *g* for 10 min, and the supernatant was collected. After measuring the protein concentration with a protein assay kit (Bio-Rad, Hercules, CA, USA), a proper amount of protein was mixed with SDS sample buffer, separated by SDS-polyacrylamide gel electrophoresis, transferred to a polyvinylidene difluoride membrane (Bio-Rad) and blocked overnight with 0.1% Tween 20 and 5% skim milk in PBS. The membranes were then incubated with the primary antibody for 1 h at room temperature or overnight at 4 °C. The membranes were washed thrice with 0.1% Tween 20 in TBS and stained with horseradish peroxidase–conjugated secondary antibodies. All immunoblots were detected by the enhanced chemiluminescence system (Amersham, Buckinghamshire, UK) according to the manufacturer’s instructions.

### 4.5. Chromatin Immunoprecipitation (ChIP) Assay

A ChIP assay was performed with an EpiXplore^TM^ ChIP Assay Kit: Anti-mouse IgG (Takara, Kyoto, Japan) as described in the manufacturer’s instructions. Cells were fixed with 1% formaldehyde to cross-link the proteins and DNA, after which DNA was sheared to fragments ranging from 200 to 800 bps by sonication in an ultrasound bath with ice. The chromatin was then incubated and precipitated with anti-HIF-1α antibody (Abcam, Cambridge, UK) and magnetic beads conjugated with anti-mouse IgG. A certain chromatin was incubated and purified without antibody as an input at the same condition too. The purified immunoprecipitated DNA fragments were detected using a Blend Taq-Plus PCR kit (Toyobo, Osaka, Japan) according to the manufacturer’s instruction. The primers coding at the VEGF promoter were designed as follows: forward (5′-TCTTTAGCCAGAGCCGG-3′) and reverse (5′-GGAATCCTGGAGTGACC-3′).

### 4.6. Fluorescence Microscope Assay

The pTK-HIF-1α-EGFP-N1 plasmid was prepared by conjugating full-length human HIF-1α cDNA to the pTK-EGFP-N1 vector described in a previous paper [16], which contained the TK promoter, according to the manufacturer’s protocol. In brief, HIF-1α cDNA with a full sequence was obtained by RT-PCR with paired primers (5′-GCTAGCATGGAGGGCGCCG-3′ containing the Nhe I site and 5′-GCTAGCCGATTCACCATGGAGGGCG-3′ containing the BamH I site). The amplified HIF-1α cDNA sequence was cloned into the pTK-EGFP-N1 vector to create a pTK- HIF-1α-EGFP-N1 plasmid containing the TK-promoter and able to express the EGFP fluorescent protein. Transfection of HCC cells with the plasmid was performed using Lipofectamine LTX (Invitrogen, Carlsbad, CA, USA) according to the manufacturer’s protocol. At 20 h to 24 h after transfection, the cells were reseeded to 3.5-cm dishes, cultured for another 24 h and treated with or without TPA and/or VK2 in combination with the prior treatment of siRNAs against PKC isoforms before being observed with a fluorescence microscope (FLoid Cell Imaging Station; Life Technologies). The cells expressing HIF-1α-EGFP in the nucleus were considered positive, and Hoechst nuclear staining was used as the control to confirm the nuclear localization.

### 4.7. SiRNA-Mediated Knockdown of PKC

The Huh7 cells were cultured and maintained in DMEM with 10% FBS and 1% ampicillin and incubated until 80% confluent at 37 °C. The cells were then replaced with new DMEM containing 10% FBS without antibiotics after being washed twice with DMEM, followed by the addition of OPTI-MEM Reduced Medium (Life Technologies) containing siRNA and Lipofectamine RNAiMAX (Life Technologies) complex, according to the manufacturer’s protocol. After incubation, the transfected cells were used for subsequent treatments and experiments.

### 4.8. Statistical Analyses

Data is shown in the text and manuscript as mean ± SD. The results of qRT-PCR were shown as fold-induction. Distribution and linearity of data were checked, differences were analyzed using Student’s *t*-test, and *p* < 0.05 was considered significant. All experiments were repeated at least three times.

## 5. Conclusions

We showed that PKC stimulation increased the HIF-1α transcriptional activity through PKC-δ by controlling the protein expression and translocation of HIF-1α to the nucleus. VK2 was shown to inhibit both NF-κB activation and PKC activity and revealed the inhibitory effects on HIF-1α transactivation. Since both HIF-1 and NF-κB, as well as PKCs, may be promising targets in cancer therapy, VK2 might be useful for the prevention and treatment of cancer, including HCC.

## Figures and Tables

**Figure 1 ijms-20-01022-f001:**
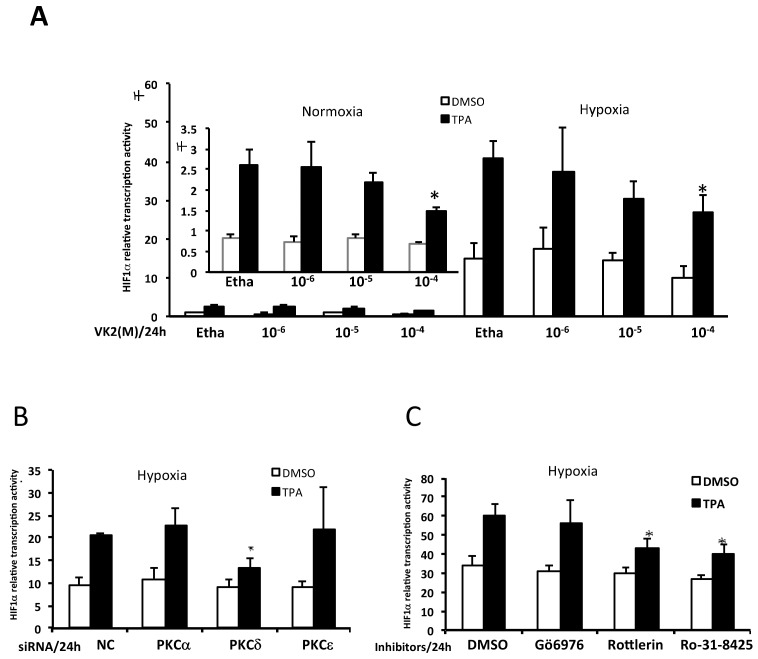
VK2 inhibited TPA-induced HIF-1α-driven luciferase expression, and PKC-δ is involved in the activation of HIF-1α transcription under both normoxic and hypoxic conditions in Huh7 cells. (**A**) The effects of VK2 on the HIF-1α-driven luciferase assay under normoxic (left and shown in enlarged panel) and hypoxic (right) conditions (*n* = 5). Hypoxia upregulated the HIF-1α luciferase activity more than 10-fold, and TPA increased the luciferase activity 3- to 4-fold both under normoxic and hypoxic conditions. VK2 dose-dependently suppressed the TPA-induced HIF-1α luciferase activity under both conditions. (**B**) The effects of PKC isoform knockdown by specific siRNAs on the HIF-1α transcriptional activity (*n* = 5). Knockdown of PKC-δ inhibited the TPA-induced HIF-1a luciferase activity under hypoxic conditions, whereas that of PKC-α or PKC-ε showed no marked effects. Without TPA, no significant changes were induced by any PKC isoform siRNAs. (**C**) The effects of PKC inhibitors on the HIF-1α transcriptional activity (*n* = 5). The PKC-δ inhibitor rottlerin (10 nM) significantly inhibited the TPA-induced HIF-1α luciferase activity to the same degree as the pan-PKC inhibitor Ro-31-8425 (100 nM). The PKC-α inhibitor Gö6976 (10 nM) did not show any suppressive effects. Data were obtained from at least three independent experiments. Bars, standard deviation; * *P* < 0.05 (Student′s *t*-test). Etha, Ethanol; NC, negative control siRNA.

**Figure 2 ijms-20-01022-f002:**
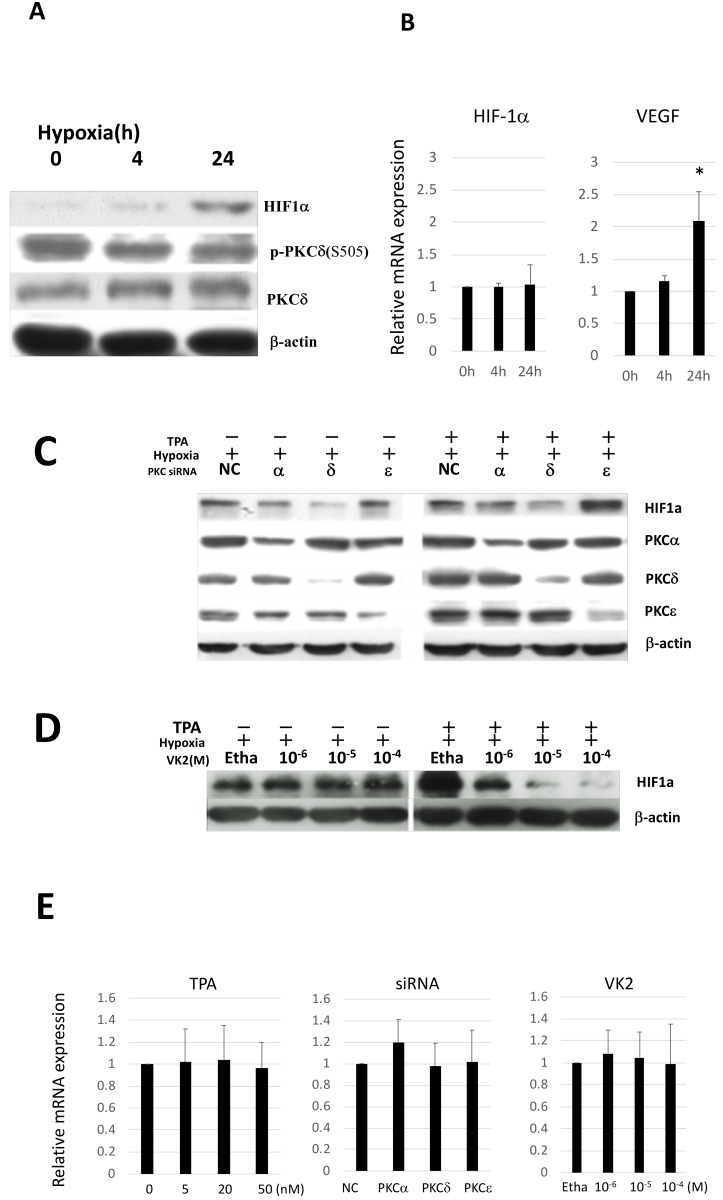
The effects of hypoxia, TPA, PKC isoforms and VK2 on the expression of HIF-1α mRNA and protein were examined in Huh7 cells. (**A**) The expression of HIF-1α protein was induced under hypoxic conditions. The phosphorylation of the PKC-δ A-loop site (S505) and total PKC-δ expression were not affected. (**B**) VEGF mRNA was induced by hypoxia, whereas the HIF-1α mRNA expression was unchanged (*n* = 4). (**C**) Knockdown of PKC-δ inhibited the expression of HIF-1α protein under hypoxic conditions, regardless of TPA induction. (**D**) VK2 suppressed the HIF-1α protein expression induced by TPA in a dose-dependent manner under hypoxic conditions, while no marked effect was observed under hypoxic conditions without TPA stimulation. (**E**) The effects of TPA, siRNAs of PKC isoforms and VK2 on the HIF-1α mRNA expression under hypoxic conditions (*n* = 4). These treatments did not alter the expression of HIF-1α mRNA. Ctr, no treatment; Etha, Ethanol; NC, negative control siRNA.

**Figure 3 ijms-20-01022-f003:**
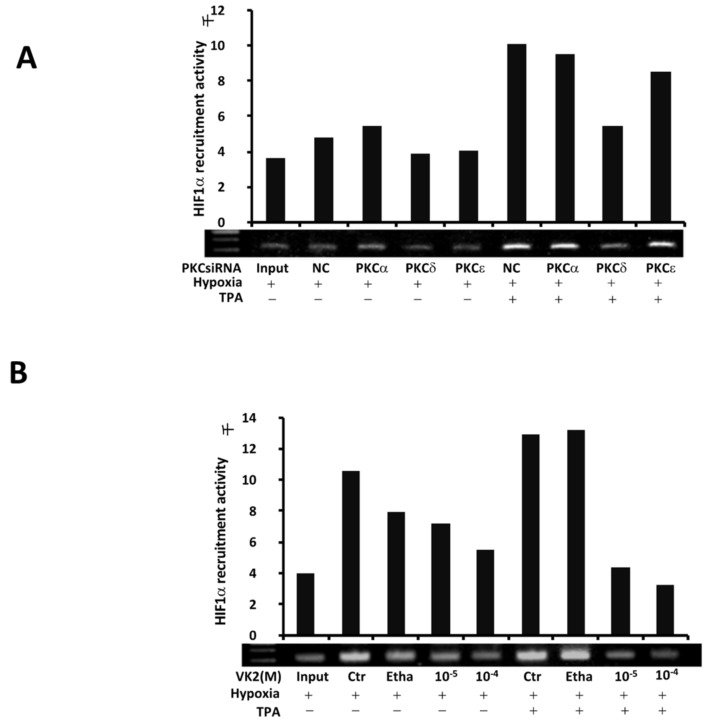
PKC-δ controlled the TPA-induced recruitment of HIF-1α to the VEGF promoter region, and VK2 abrogated the induction of HIF-1α recruitment by TPA in Huh7 cells. After treatments, the cells were harvested immediately and subjected to subsequent ChIP assays as described in the Materials and Methods. (**A**) The effects of PKC isoforms on TPA-induced HIF-1a recruitment. Cells were treated with isoform-specific PKC siRNAs under hypoxic conditions, and a ChIP assay was performed immediately after treatment. Only PKC-δ knockdown abrogated the TPA-induced recruitment of HIF-1α. (**B**) VK2 abrogated the effects of TPA on the HIF-1a recruitment activity under hypoxic conditions in Huh7 cells. Ctr, no treatment; Etha, Ethanol; NC, negative control siRNA.

**Figure 4 ijms-20-01022-f004:**
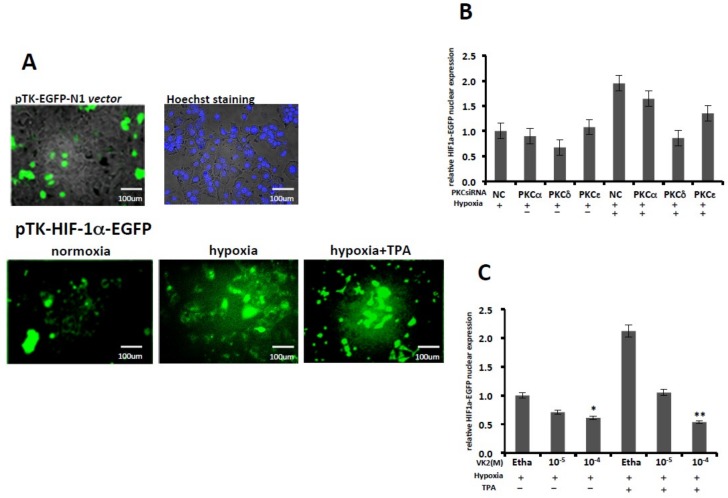
The role of PKC isoforms and the effects of VK2 on the nuclear translocation of HIF-1α in Huh7 cells. After transfection with pTK-EGFP-N1 or pTK-HIF-1α-EGFP-N1 plasmids, the cells were plated onto 3.5-cm dishes for further treatment and subjected to observation under a microscope. (**A**) Upper panel: the pTK-EGFP-N1 vector expression under normoxic conditions in Huh7 cells. Nuclear staining by Hoechst was done as the control. Lower panel: the pTK-HIF-1α-EGFP-N1 vector expression under the indicated conditions. (**B**) The effects of PKC isoform knockdown on the translocation of HIF-1α induced by TPA under hypoxic conditions. The HIF-1α-EGFP expression was observed with Hoechst staining as a control under florescence microscope. The positive cells were counted and totaled in the graph. (**C**) The effects of VK2 on the translocation of HIF-1α-EGFP to the nucleus. VK2 inhibited the nuclear expression of EGFP-positive cells. Thousands of cells were counted in each experiment. Experiments were repeated three times, and similar results were obtained. * *p <* 0.05; ** *p <* 0.01 (Student′s *t*-test). Etha, Ethanol; NC, negative control siRNA.

**Figure 5 ijms-20-01022-f005:**
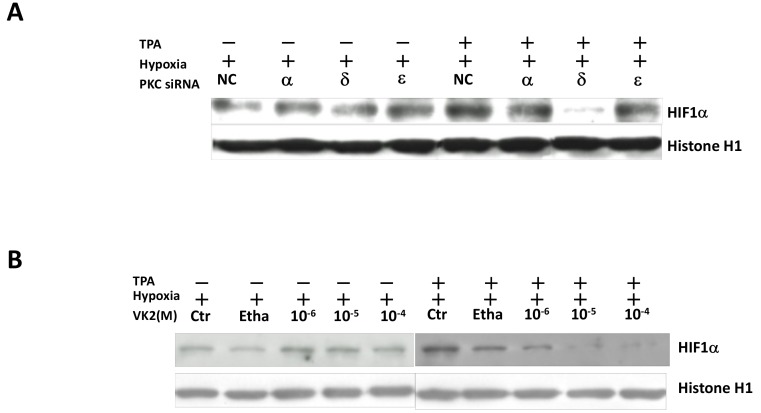
The role of PKC isoforms and the effects of VK2 on the nuclear accumulation of HIF-1α protein under hypoxic conditions in Huh7 cells. (**A**) TPA induced the nuclear expression of HIF-1a protein in Huh7 cells under hypoxic conditions. The increase in the HIF-1α protein expression in the nucleus was largely suppressed by the knockdown of PKC-δ not by that of PKC-α or PKC-ε. The cells were treated with PKC siRNAs and incubated for 24 h under hypoxic conditions with or without 50 nM TPA. After treatments, the nuclear proteins were extracted and subjected to Western blotting with specific antibodies, as indicated. Histone H1 was used as nuclear protein loading control. (**B**) VK2 inhibited the nuclear accumulation of HIF-1α induced by TPA in a dose-dependent manner in Huh7 cells under hypoxic conditions.

**Figure 6 ijms-20-01022-f006:**
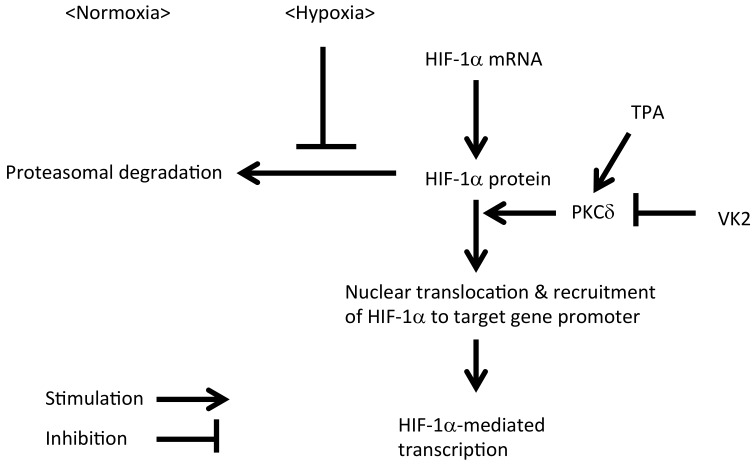
A proposed signaling pathway that regulate HIF-1α transcription activity through PKC-δ isoform and VK2.

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
