# Peer review of "Mechanisms of PKC-Mediated Enhancement of HIF-1α Activity and its Inhibition by Vitamin K2 in Hepatocellular Carcinoma Cells"

_ijms, 2019, doi:10.3390/ijms20051022_

Round 1
Reviewer 1 Report
Maybe a little clinical touch shouild be added. The dose and type of vitamin K2 is important. Previous studies have often used MQ-4 whereas recent studies focus om MQ-7 in higher doses and intake over longer periods to evaluate extrahepatic effects on osteocalcin, Matrix-Gla-Protein and Gas6. Very few adequately powered studies on vitamin K2 supplementation or indirect calculations from queries on vitamin K2 food intake have high evidence levels on reduced cancer growth, protection against metastases, etc... In HCC, PIVKA-II analyses have béen used to prognostizise HCC outcome - also trying to reduce it with better food or vitamin supplementation. However the PIVKA-II rise in HCC does not depend on vitamin K deficiency. I would like the authors to adress these suggestions.
In methods: I would like to have a better description how vitamin K2 treatment/incubation of the HCC-cells were performed.
Also the origin of HCC could affect the outcome- hepatitis origin or not? Comment
Author Response
Answers to Reviewer1
>Maybe a little clinical touch shouild be added. The dose and type of vitamin K2 is important. Previous studies have often used MQ-4 whereas recent studies focus om MQ-7 in higher doses and intake over longer periods to evaluate extrahepatic effects on osteocalcin, Matrix-Gla-Protein and Gas6. Very few adequately powered studies on vitamin K2 supplementation or indirect calculations from queries on vitamin K2 food intake have high evidence levels on reduced cancer growth, protection against metastases, etc... In HCC, PIVKA-II analyses have béen used to prognostizise HCC outcome - also trying to reduce it with better food or vitamin supplementation. However the PIVKA-II rise in HCC does not depend on vitamin K deficiency. I would like the authors to adress these suggestions.
According to the reviewer’s suggestion, we added the discussion regarding PIVKA-II in HCC diagnosis and clinical use of vitamin K in HCC patients providing references. We also discussed about the recent study about use of menaquinone-7 (MK-7) in extrahepatic tissue. All the modified portion are underlined.
>In methods: I would like to have a better description how vitamin K2 treatment/incubation of the HCC-cells were performed.
According to the reviewer1’s suggestion, we added the procedure of vitamin K2 treatment of HCC cells in the section of materials and methods.
>Also the origin of HCC could affect the outcome- hepatitis origin or not?
Most of the studies used HCC cases with viral hepatitis including hepatis C or hepatitis B. There are no data of VK2 treatment based on etiology. Therefore outcome based on origin of HCC are not known at this point.
Reviewer 2 Report
In the manuscript entitled "Mechanisms of PKC-mediated enhancement of HIF-1α activity and its inhibition by vitamin K2 in hepatocellular carcinoma cells", Xia J and collaborators have explored the putative role of PKC isoforms in the activation of HIF-1a in human HCC cells and the effects of vitamin K2 on HIF-1a activation, using a wide range of molecular biology techniques to reinforce their findings. The results derived from this manuscript not only establish a role between PCK-d and HIF-1a, but also a regulatory role of vitamin K2 on HIF-1a activation. Even thought the manuscript is well-written and includes a proper explanation of background, methods, results and their discussion, some modifications should be taken into account before it is considered for publication.
Please, consider using these abbreviations:
TF, transcription factor
TPA, 12-O-tetradecanoylphorbol-13-acetate
Minor changes:
"3. Discussion section": please, consider the inclusion of an image/figure with a summary of the results of the paper. This image may illustrate the link between the modulated pathways evaluated in this manuscript.
Line 224: "Cells were cultured..."
Lines: 231-232: Sigma.Aldrich (City, Country). Please, ensure that all the compounds and equipment includes city and country.
Line 239: "Luciferase reporter gene assay"
Line 243: 2x103 cells per well
Line 248: "Cell extracts..."
Line 251: "Pierce 660nm protein assay system"
Line 254: Real-Time qRT-PCR, real-time quantitative reverse transcription polymerase chain reaction.
Line 276: "at 4 ºC"
Line 282: Should be "4.5 Chromatin..."
Line 295: "4.6. Fluorescence..."
Section "2.8 Statistical analyses": Please add: "Data is shown in the text and manuscript as mean ± SD. Based on the number of replicates, authors should state if the checked for normality of the variables analyzed, if they performed any transformation of the data to normalize data, and evaluate the convenience of using a non-parametric test (e.g. Wilcoxon) when appropriate.
Figures
As figures must stand alone, please, include the definition of all the abbreviations used together with the meaning of the legend (e.g. *).
Please, include in all the legends the number of replicates (technical and/or biologicals) that you have used for each outcome.
I suggest removing the lines linking the comparisons in the figures and state in the figure legend that "*, P<0.05, Student's t-test versus the XX". In case of Fig 1.A: "*, P<0.05, Student's t-test versus the Etha condition".
Author Response
Answers to Reviewer2
>In the manuscript entitled "Mechanisms of PKC-mediated enhancement of HIF-1α activity and its inhibition by vitamin K2 in hepatocellular carcinoma cells", Xia J and collaborators have explored the putative role of PKC isoforms in the activation of HIF-1a in human HCC cells and the effects of vitamin K2 on HIF-1a activation, using a wide range of molecular biology techniques to reinforce their findings. The results derived from this manuscript not only establish a role between PCK-d and HIF-1a, but also a regulatory role of vitamin K2 on HIF-1a activation. Even thought the manuscript is well written and includes a proper explanation of background, methods, results and their discussion, some modifications should be taken into account before it is considered for publication.
>Please, consider using these abbreviations:
TF, transcription factor. Added to abbreviations
TPA, 12-O-tetradecanoylphorbol-13-acetate. Added to abbreviations
Minor changes:
> "3. Discussion section": please, consider the inclusion of an image/figure with a summary of the results of the paper. This image may illustrate the link
between the modulated pathways evaluated in this manuscript.
According to the reviewer’s suggestion, we summarized the results as new Figure 6 showing the link that regulate HIF-1a transcriptional activity.
> Line 224: "Cells were cultured..."
Corrected
>Lines: 231-232: Sigma.Aldrich (City, Country). Please, ensure that all the compounds and equipment includes city and country.
According to the reviewer’s suggestion we have checked the compounds and equipment. We described the company name with city and country when first appeared in the text.
>Line 239: "Luciferase reporter gene assay"
Corrected.
>Line 243: 2x103 cells per well
Corrected.
>Line 248: "Cell extracts..."
Corrected.
>Line 251: "Pierce 660nm protein assay system"
Corrected.
>Line 254: Real-Time qRT-PCR, real-time quantitative reverse transcription
polymerase chain reaction.
Corrected.
>Line 276: "at 4 ºC"
Corrected.
>Line 282: Should be "4.5 Chromatin..."
Corrected.
Line 295: "4.6. Fluorescence..."
Corrected.
>Section "2.8 Statistical analyses": Please add: "Data is shown in the text and manuscript as mean ± SD. Based on the number of replicates, authors should state if the checked for normality of the variables analyzed, if they performed any transformation of the data to normalize data, and evaluate the convenience of using a non-parametric test (e.g. Wilcoxon) when appropriate.
According to the reviewer’s suggestion, we added the text” Data is shown in the text and manuscript as mean ± SD.” The results of luciferase assay and qRT-PCR data are shown as fold induction. We checked the distribution and linearity of data and used Student’s t-test.
>Figures
As figures must stand alone, please, include the definition of all the
abbreviations used together with the meaning of the legend (e.g. *).
Please, include in all the legends the number of replicates (technical and/or
biologicals) that you have used for each outcome.
I suggest removing the lines linking the comparisons in the figures and state
According to the reviewer’s suggestion, we added the definition of abbreviations in the figure legends and the number of replicates. We also removed the lines of Fig.1. All the modified portion of text were underlined in the revised version.